# A Functional Pipeline of Genome-Wide Association Data Leads to Midostaurin as a Repurposed Drug for Alzheimer’s Disease

**DOI:** 10.3390/ijms241512079

**Published:** 2023-07-28

**Authors:** Alvaro Esteban-Martos, Ana Maria Brokate-Llanos, Luis Miguel Real, Sonia Melgar-Locatelli, Itziar de Rojas, Adriana Castro-Zavala, Maria Jose Bravo, Maria del Carmen Mañas-Padilla, Pablo García-González, Maximiliano Ruiz-Galdon, Beatriz Pacheco-Sánchez, Rocío Polvillo, Fernando Rodriguez de Fonseca, Irene González, Estela Castilla-Ortega, Manuel J. Muñoz, Patricia Rivera, Armando Reyes-Engel, Agustin Ruiz, Jose Luis Royo

**Affiliations:** 1Department of Surgery, Biochemistry and Immunology, School of Medicine, University of Malaga, Boulevard Louis Pasteur s/n, 29071 Malaga, Spain; alvaroesteban@uma.es (A.E.-M.); lmreal67b@gmail.com (L.M.R.); mjbravo@uma.es (M.J.B.); mruizg@uma.es (M.R.-G.); igr@uma.es (I.G.); engel@uma.es (A.R.-E.); 2Departamento de Biología Molecular e Ingeniería Bioquímica, Centro Andaluz de Biología del Desarrollo (CABD), Universidad Pablo de Olavide (UPO), UPO/CSIC/JA, Ctra Utrera Km1, 41013 Sevilla, Spain; ambrolla@upo.es (A.M.B.-L.); rpolher@upo.es (R.P.); mmunrui@upo.es (M.J.M.); 3Centro de Investigación Biomédica en Red de Enfermedades Infecciosas (CIBERINFEC), 28029 Madrid, Spain; 4Instituto de Investigación Biomédica de Málaga y Plataforma en Nanomedicina-IBIMA Plataforma BIONAND, 29590 Malaga, Spain; soniaml1998@uma.es (S.M.-L.); adriana.castro@uma.es (A.C.-Z.); mmp241@uma.es (M.d.C.M.-P.); beatriz.pacheco@ibima.eu (B.P.-S.); fernando.rodriguez@ibima.eu (F.R.d.F.); estela.castilla@ibima.eu (E.C.-O.); patricia.rivera@ibima.eu (P.R.); 5Departamento de Psicobiología y Metodología de las Ciencias del Comportamiento, Facultad de Psicología, Universidad de Málaga, 29071 Malaga, Spain; 6Research Center and Memory Clinic, Ace Alzheimer Center Barcelona—Universitat Internacional de Catalunya, 08017 Barcelona, Spain; iderojas@fundacioace.org (I.d.R.); pgarcia@fundacioace.org (P.G.-G.); aruiz@fundacioace.org (A.R.); 7Network Center for Biomedical Research in Neurodegenerative Diseases (CIBERNED), National Institute of Health Carlos III, 28029 Madrid, Spain; 8Unidad de Gestion Clinica de Salud Mental, Hospital Universitario Regional de Malaga, 29010 Malaga, Spain

**Keywords:** Alzheimer’s disease, GWAS pipeline, drug discovery, midostaurin

## Abstract

Genome-wide association studies (GWAS) constitute a powerful tool to identify the different biochemical pathways associated with disease. This knowledge can be used to prioritize drugs targeting these routes, paving the road to clinical application. Here, we describe DAGGER (Drug Repositioning by Analysis of GWAS and Gene Expression in R), a straightforward pipeline to find currently approved drugs with repurposing potential. As a proof of concept, we analyzed a meta-GWAS of 1.6 × 10^7^ single-nucleotide polymorphisms performed on Alzheimer’s disease (AD). Our pipeline uses the Genotype-Tissue Expression (GTEx) and Drug Gene Interaction (DGI) databases for a rational prioritization of 22 druggable targets. Next, we performed a two-stage in vivo functional assay. We used a *C. elegans* humanized model over-expressing the Aβ_1-42_ peptide. We assayed the five top-scoring candidate drugs, finding midostaurin, a multitarget protein kinase inhibitor, to be a protective drug. Next, 3xTg AD transgenic mice were used for a final evaluation of midostaurin’s effect. Behavioral testing after three weeks of 20 mg/kg intraperitoneal treatment revealed a significant improvement in behavior, including locomotion, anxiety-like behavior, and new-place recognition. Altogether, we consider that our pipeline might be a useful tool for drug repurposing in complex diseases.

## 1. Introduction

Drug discovery comprises a variety of scientific fields, including pharmacology, biochemistry, and molecular biology. A central aim of drug discovery initiatives is to identify effective and safe molecular entities useful in the daily management of disease. The entire procedure of drug discovery is a very complicated, time-consuming, expensive, and financially risky effort. Neurodegenerative diseases constitute some of the greatest challenges for the pharmaceutical industry. This is especially relevant for Alzheimer’s disease (AD), the main worldwide cause of dementia [1,2,3,4]. AD has an estimated heritability of 58% to 76%, pinpointing the importance of interindividual genetic differences in its development [5]. Depending on the age of appearance, two different forms of AD can be found, largely depending on the model of inheritance. Early-onset Alzheimer’s disease is diagnosed before the age of 60 and is associated with the presence of highly penetrant but low-prevalent mutations such as those mapping *APP*, *PSEN1*, or *PSEN2* genes [6]. However, most cases are categorized as late-onset AD, whose appearance has been associated with low-penetrance but highly prevalent polymorphisms [7]. In previous decades, drugs approved by the Food and Drug Administration (FDA) and European Medicines Agency (EMA) have contributed to symptom alleviation in patients rather than to a clear improvement of cognitive function [8]. More recently, anti-amyloid beta (Aβ) monoclonal antibodies have been developed. First, aducanumab and, shortly after, lecanemab showed a significant reduction of Aβ levels in AD patients, which have been shown to slow down cognitive decay. However, the efficacy of these therapies has been questioned [9,10,11,12], and the recently FDA-approved lecanemab phase IV trials are not yet available. Furthermore, although the effect of Aβ clearance has been associated with a significant reduction in cognitive decline, there is still a margin for significant improvement. This leaves room for improvement. Perhaps additional intervening pathways can eventually find a synergic therapeutic effect. We should outline that the involvement of Aβ in AD pathogenesis has been known since 1984. This highlights that the development of effective treatments for AD, as for other complex diseases, is a very slow process. This can be attributed to multiple factors, such as the problems in identifying key druggable targets, the high costs of drug development, potential pharmacokinetic problems, insufficient permeability across the blood–brain barrier, and many others. Furthermore, once preclinical studies are completed, side effects may be found in clinical trials. These obstacles can be avoided with the strategy of drug repurposing, which studies the effectiveness of drugs, already approved by the FDA and/or the EMA, in new clinical applications [13,14]. In order to design a repurposing strategy, we first need to identify the biochemical pathways that lie underneath the pathophysiology of the studied disease. These studies typically use the genotyping of patients and healthy controls for genome comparison, known as genome-wide association studies (GWAS). Over the past two decades, GWAS have allowed the scientific community to find both risk and protective factors affecting complex trait etiology. Recently, the largest meta-GWAS in AD was published by the European Alzheimer’s Disease DNA BioBank (EADB) Consortium [15]. The results of this study, performed in LOAD patients, constitute a remarkable source of genetic information that can be translated to tissue-specific gene expression and used for drug discovery. In this paper, we present a new pipeline for drug repurposing for AD based on this meta-analysis. Although extremely helpful in the identification of the biochemical pathways involved in the development of AD, GWAS do not allow a full understanding of the molecular etiology behind genetic diseases, given the lack of functional information. For this reason, integrating gene expression data is crucial to understanding the impact of the epidemiologically associated polymorphisms on cellular function. DAGGER (Drug Repositioning by Analysis of GWAS and Gene Expression in R) combines the power of GWAS with the functional approach of transcriptomics. GWAS signals were filtered according to the transcriptomic impact and queried in the Drug Gene Interaction Database (DGIdb), which contains experimentally confirmed information on commercially available drugs [16].

## 2. Results

The meta-GWAS used for target prioritization involved 16,358,691 single-nucleotide polymorphisms (SNPs) from 111,325 AD patients and 677,663 healthy controls [15]. As a first step in the analysis, we defined a low-stringency *p*-value cutoff of 5 × 10^−3^, resulting in 41,919 SNPs. A set of random SNPs of equal size was analyzed simultaneously. The control series showed a random distribution along the human genome, while in the associated SNP distribution, we could observe that chromosomes 6, 11, 17, and 19 concentrated more mutations than in the random selection, as expected from previous results [17,18]. Volcano plots of the sets showed symmetry in both risk and protective signals (Appendix A). As expected, the significant volcano *p*-value plot reaches much lower scores than the random plot.

Next, we outcrossed the SNP list with the GTEx database (Figure 1), selecting variants significantly associated with changes in mRNA levels (eQTLs). A total of 439 of the associated SNPs resulted in statistically significant eQTLs. This was a considerably higher proportion than what was obtained with the random SNP selection (*n* = 123) (1.2% vs. 0.38%, *p* = 1.50 × 10^−40^, χ^2^ test). Randomly selected eQTLs were removed from the significant dataset, resulting in a final selection of 435 eQTLs affecting the expression of 298 genes. We defined the proteins encoded by these genes as potential targets for drug repositioning. Enrichment analysis revealed that all pathways identified were previously associated with AD (Appendix A). No significant enrichment was found in the network generated from the random dataset. A summary of the results obtained with the DAGGER pipeline can be found in Appendix A.

Next, we compared our gene association strategy with the results obtained using a previously reported pipeline named gprofiler2. This R package links the candidate SNP to the nearest gene. When both GWAS SNP lists (significant and random) were introduced in gprofiler2, the associated genes were compared with those obtained using DAGGER. Interestingly, DAGGER and gprofiler2 differed in 95% of the associated genes which highlights the difference between structural and functional approaches to SNP–gene association. DAGGER’s full results can be found in Appendix A. Gprofiler2 results on the significant and random datasets can be found in Appendix A, respectively.

When the 298 DAGGER candidates were compared to the ADSP Gene Verification Committee database (NG00076), containing 20 confirmed AD-associated genes and 76 *loci* with genetic evidence of AD implication, 5 matches were found. These genes belong to the genetic evidence group, and they are *ACE*, *APH1B*, *PTK2B*, *INPP5D*, and *PRKD3*. This suggests that our analysis identified 278 additional *loci* potentially associated with the disease.

Next, we addressed the molecular interactions between the available drugs and the targets, taking into account both epidemiological and expression data. For instance, a mutation increasing AD risk associated with an augmented gene expression led us to search for an inhibitor. This would mimic, regardless of the patient’s genotype, the beneficial effect of the mutation. On the contrary, protective mutations associated with gene over-expression would lead us to search for an agonist. Thus, the candidate list was outcrossed with the information available on the DGIdb. On this first run of DAGGER, 69 out of the 298 identified targets were identified as pharmacologically intervenable with EMA- and/or FDA-approved drugs. From these, 22 candidates fit the aforementioned SNP–drug interaction criteria. Out of this selection, we prioritized those genes whose expression levels were especially relevant in the central nervous system (CNS) (Table 1). However, we discarded *MARK3* from further studies since its involvement in AD has been already described [19].

We then selected the best candidates for a *C. elegans* in vivo functional assay. We directly found orthologs in the Ensembl database for *DMPK*, *KCNN4*, and *NDUFS2*. We also analyzed the rest of the candidates with BLASTp against the *C. elegans* proteome. Several matches were found for *PRKD3* in *C. elegans* with identities ranging from 53.52% to 65.83%, which were included in the candidate list. We should also take into account that GTEx project donor availability largely depends on the tissue of interest, and brain-derived samples often have a lower sample size when compared to other tissues. Some of the identified SNPs were eQTL on highly accessible tissues but not statistically significant in CNS due to low statistical power. This was the case for *KCNH6*, and for this reason, we manually included it for further characterization. With this incorporation, we reached a final candidate selection list shown in Table 2. The full DAGGER results table can be found in Appendix A.

These five selected drugs were assayed at clinical range concentrations described in the European Medical Agency medication datasheets. We first used the humanized *C. elegans* GMC101 strain, where full-length human Aβ_1-42_ is expressed in muscles. This strain generates amyloid aggregates when incubated at 25 °C, which correlates with paralysis [20,21,22,23]. NS1643 resulted in worm developmental problems and was discarded from the study. Cox regression analysis of the motility curves assaying NV-128, RKI-1477, and clotrimazole resulted in a significantly worse fitness of the animals when compared with untreated ones. However, midostaurin resulted in a significant increase in worm motility (*p* = 0.036, Mantel–Cox test) (Figure 2, Appendix A).

In the final stage of our pipeline, midostaurin was assayed in the 3xTg AD mouse model. Six-month 3xTg mice were subjected to either placebo or 20 mg/kg midostaurin with daily intraperitoneal administration for 3 weeks before behavioral evaluation (Figure 3A). We observed that treated mice were more active. In the elevated plus maze test, treated mice showed an increased, although not significant motility (average of 930 vs. 1203 cm; vehicle and treatment, respectively). In addition, treated mice were more prone to explore the open arm, suggesting lower anxiety-like levels (11 vs. 39 s, *p* < 0.05, Student’s *t*-test) (Figure 3B). Results from the elevated plus maze replicated the previous motility results since treated mice showed nearly 40% increased locomotion when compared to those treated with vehicle (1880 vs. 1350 cm; *p* < 0.01, Student’s *t*-test). When the center exploration time was analyzed, treated mice showed higher times (33 vs. 53 s), although these results did not reach statistical significance (Figure 3C). Regarding memory tests, NOR values of treated animals were higher, although not statistically significant, than those obtained from the animal with vehicle (average 0.07 vs. 0.21; *p* = 0.877, Student’s t-test) (Figure 3D). Finally, when NPR values were analyzed, midostaurin-treated mice were more prone to spend time exploring the new object (average 4.83 vs. 3.46 s; *p* < 0.05, Student’s *t*-test) (Figure 3E).

## 3. Discussion

Here, we describe a drug repurposing strategy based on the rational use of GWAS data. By incorporating the GTEx database into GWAS meta-analysis, DAGGER goes beyond studying mutations in specific genes, instead focusing on the functional interactions between genomics and transcriptomics. It is also not limited to known interactions, as the program itself infers the gene–transcript relationship by comparing GWAS and expression data. Other bioinformatic workflows have already exploited omics data for drug repositioning, but mainly assigning the SNP to the nearest gene [24,25]. This strategy might lead to a false interpretation of the true impact of polymorphism on gene function. Moreover, from a merely structural point of view, we cannot model whether the variant might be associated with a gain or a loss of function.

As a proof of concept, we applied our pipeline to the GWAS results previously reported by the EADB Consortium [15]. Our analysis began with the selection of the genetic signals with the highest statistical significance, which were outcrossed with GTEx data. This allowed us to find a subset of *loci* epidemiologically linked to AD and with experimental evidence of a differential expression. Our rationale included the evaluation of the odds ratio in combination with the GTEx-reported expression slope in order to predict desirable drug interactions. Drugs known to affect the proteins of interest in the desired manner were selected from the DGIdb. Given the costs of rodent model assays, we first performed an in vivo validation assay as a filter using the transgenic *C. elegans* GMC101 strain, which expresses Aβ_1-42_ after a temperature rise, leading to gradual paralysis. Our data showed a reproducible and statistically significant increase in motility when midostaurin was administered. Therefore, it was assayed in vivo using the 3xTg mice. Upon a three-week IP administration, behavioral results showed that midostaurin improved motility, reduced anxiety, and enhanced particular cognitive functions such as novel-place recognition. Midostaurin is an orally delivered, reversible, multi-targeted protein kinase inhibitor with anticancer effects. Beyond PRKD3 antagonism, midostaurin also inhibits PKCα/β/γ, Syk, Flk-1, Akt, PKA, and c-Kit, among others. Some of these proteins have never been related to a cognitive phenotype (for instance PRKD3), while others, such as PKC or Syk, have been previously associated with memory and behavior [26,27]. Thus, we cannot rule out that the true effect of midostaurin might be attributed to its pleiotropic effect over the different kinases. Eventually, specific protein kinase inhibitors shall be assayed to determine whether a more efficient intervention against PRKD3 would improve the observed phenotype. Nevertheless, the observed effects of midostaurin in AD model mice provide a promising candidate for repositioning.

A similar program, PrediXcan (current version 0.9.1), also tries to extract new information by integrating GWAS with transcriptomics and has been used in AD studies [28,29]. While PrediXcan predicts expression levels, DAGGER uses confirmed experimental data from the GTEx database. SMR [30] applies the same concept, although in a much more complex manner, making DAGGER a computationally lighter alternative. Our pipeline is currently a proof of concept, which is why it takes the simplest approach by simply selecting SNP below a certain *p*-value threshold. This constitutes a major limitation of our approach, but it has not prevented us from identifying a very promising new candidate. Future versions of DAGGER will only expand on its current capabilities, such as by taking into account the network of gene–gene interactions not only in a direct manner but also in second and eventually third orders of interaction, which could provide even more potential repositioning targets.

The DAGGER pipeline brings a different and high-throughput approach to drug repurposing. In view of the promising results described in this work, we believe that our method will significantly contribute to not only AD research but also complex genetic disease as a whole.

## 4. Materials and Methods

### 4.1. GWAS, eQTL, and Drug Targeting Analysis

GWAS summary statistic data were analyzed using the DAGGER (Drug Repositioning by Analysis of GWAS and Gene Expression in R) pipeline (version 0.9), available at https://github.com/AEstebanMar/DAGGER-publication- (accessed on 22 April 2023). Preprocessing of the data was performed in GNU/Linux 4.4.0, distribution Ubuntu 20.04.3 LTS. DAGGER code was written under R version 4.2.1 [31], using the Stringr [32] and ggplot2 [33] packages (versions 1.5.0 and 3.4.2, respectively). Inputs used were the meta-GWAS results from EADB [15], version 8 of GTEx [34], tissue-eQTL analysis (eGene and significant variant–gene associations), the interactions dataset from DGIDb [16], and the Alzforum Therapeutics Alzheimer Disease database [35]. We selected GWAS SNPs with a value of *p* = 0.001 or lower as significant. From the GTEx database, the top 5% Q-value SNP–gene expression associations were selected. According to information available at the UCSC genome browser (https://genome.cshlp.org/content/12/6/996.abstract, accessed on 22 April 2023), the Q-value is defined as the minimum false discovery rate at which an observed score is deemed significant. The Alzheimer’s Disease Sequencing Project (ADSP) database was consulted to identify new candidates among the list of targets identified in our analysis (NIAGADS NG00067) (https://adsp.niagads.org/gvc-top-hits-list/, accessed on (accessed on 22 April 2023)). Ensembl database release 108 [36] was used for *Caenorhabditis elegans* ortholog assignation. Alternatively, predicted orthologs with at least 50% identity were selected by BLASTp [37].

### 4.2. GO Enrichment and Drug Targeting

Network and enrichment analyses were performed with the STRING online tool [38], available at https://string-db.org/ (accessed on 22 April 2023). The network was constructed using the default STRING parameters, with a Markov cluster algorithm (MCL) inflation parameter of 3. Enrichment analysis was taken from the analysis tab of the generated network.

### 4.3. C. elegans Motility Assays

*C. elegans* GMC101 strain (dvls100 [unc-54p::A-beta-1-42::unc-54 3′-UTR + mtl-2p::GFP]), was used as the AD model. This strain expresses Aβ_1-42_ in muscle cells, leading to animal paralysis. This occurred more rapidly and more severely when Aβ_1-42_ was heat induced at 25 °C [39]. For paralysis assays, treatment started when animals were at the embryonic stage. Embryos were laid on NGM plates with and without each drug at 16 °C for 4 days to develop up to Larval Stage 4 (L4, just before adulthood). Sixty L4 worms (20 worms per 3 plates) were transferred to their respective NGM plates and incubated at 25 °C for 18–24 h where they reached young adulthood. Observation and quantification of paralysis were performed in adult animals every 2 or 3 h as previously stated [40], with slight modifications, since we considered the worms that were not able to move backward a distance equal to their body size paralyzed. The different drugs were prepared at 100× the final concentration in 100% dimethylsulfoxide (DMSO). The final concentrations in plates were midostaurin, 50 mM; clotrimazol, 0.1 μM; NS1643, 30 μM; RKI-1447, 20 μM; and NV-128, 150 μM (MedchemExpress LLC, Monmouth Junction, NJ, USA). As the negative control, 1% DMSO was used. The experiments were repeated at least twice.

### 4.4. AD Model Mice

Animal experimental procedures were carried out following the European Communities Council Directives 2010/63/EU, Regulation (EC) No.: 86/609/ECC (24 November 1986) and the Spanish National and Regional Guidelines for Animal Experimentation (Real Decreto 53/2013). Experimental protocols were approved by the Local Ethical Committee for Animal Research of the University of Malaga (CEUMA No.: 105-2020-A).

Experiments on mice were performed in the triple-transgenic mice B6;129-Psen1tm1Mpm Tg (APPSwe, tauP301L)1Lfa/J (named 3xTg) [41] purchased from the Jackson Laboratory (Bar Harbor, ME, USA) in both male and female individuals. The mice (*n* = 8 vehicle; *n* = 10 treatment), aged 6 months and weighing 20–25 g at the moment of experiments, received either 10% DMSO in physiological serum as vehicle or midostaurin 20 mg/kg/d in the treatment group for 21 consecutive days. Animals were placed in individual standard cages in a temperature- and humidity-controlled room, under a 12 h light/dark cycle, and with ad libitum access to food and water.

### 4.5. Behavioral Testing

In order to evaluate the treatment effect in the 3xTg model, we performed different behavioral tests. All of these experiments began one day after the last treatment administration. In all cases, mice were carried to a noise-isolated room (illuminated 120 lux) at 9:00 a.m. and stayed there for at least 20 min for habituation. Before each animal was tested, the maze arena was cleaned with 30% ethanol to eliminate odor cues [42,43]. In every test, sessions were recorded using an overhead digital camera and analyzed with Ethovision XT software v17 (Noldus, Wageningen, The Netherlands) for spatiotemporal and observational measures. Observational scoring was carried out by a trained observer who was blind to the mice’s sex or experimental condition and had no previous assumptions about the study’s outcome. Behavioral assessment was carried out following previously reported protocols [43,44,45,46]. On the first day of behavioral assessment (i.e., Day 22 of the experiment), we evaluated motility and anxiety-like behavior in the elevated plus maze (EPM). The apparatus consisted of two open arms (29.5 × 5 cm), two closed arms (29.5 × 5 cm), and a central connecting platform (5 × 5 cm) raised 47 cm above the floor. Each mouse was placed in the maze’s center for 5 min. Total locomotion and the time spent in the open arms were measured. On Day 23, mice were released to freely explore an empty open field (OF) (40 × 40 cm; 40 cm high) for 5 min for the assessment of locomotion and time in the center zone (defined as a central square of 20 × 20 cm). One hour after the OF exploration session, mice were subjected to the novel object recognition (NOR) task. Briefly, two identical copies of an object (i.e., “familiar” object) were placed in the apparatus near two adjacent corners, and mice were left to explore for 10 min (sample session). On Day 24, mice were allowed to explore for 10 min an identical copy of the familiar object and a “novel” unknown object located in one of the previous positions (object memory test). The preference ratio (%) [(time exploring the novel object−time exploring the familiar object)/total time exploring both objects] and time spent exploring both objects were determined. Finally, on Day 25, the novel-place recognition (NPR) test was performed. Mice were allowed to explore for 10 min two identical copies of the familiar object, one placed in its habitual position and the other displaced to an opposite corner (place memory test). The preference ratio (%) [(time exploring the displaced object−time exploring the static object)/total time exploring both objects] and total time exploring the familiar and displaced objects were determined.

### 4.6. Statistical Analysis

We tested the normality of data using Kolmogorov–Smirnov’s test, and outliers were checked by Grubbs’ test. Data from female and male mice were used for analysis, as no significant differences between sexes were found by Student’s *t*-test. For the normal single-variable case, we analyzed data using the two-tailed Student’s *t*-test and the two-tailed Mann–Whitney test for data with unpaired non-normal distribution or the Wilcoxon test for paired non-normal distribution data. *C. elegans* motility analysis was plotted using Kaplan–Meier diagrams followed by the Mantex–Cox test. GraphPad Prism 9.5.0 and IBM SPSS Statistics software v23 were used for statistical analysis and graphing. Data are represented as mean ± SE, and *p* < 0.05 was considered significant.

## Figures and Tables

**Figure 1 ijms-24-12079-f001:**
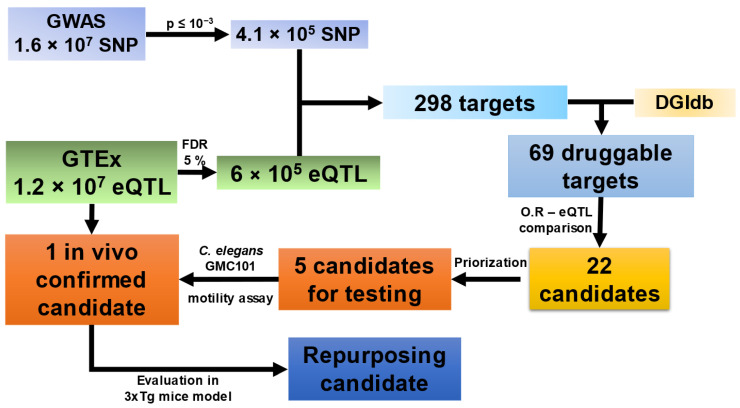
Workflow overview. Steps up to and including OR-eQTL comparison are part of the DAGGER pipeline.

**Figure 2 ijms-24-12079-f002:**
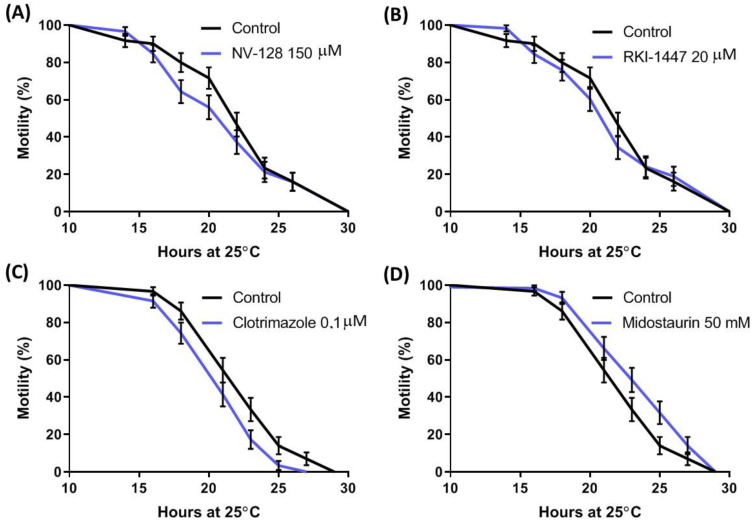
Effects on paralysis rate upon treatment with the selected drugs in populations of AD *C. elegans* model GMC101 strain: (**A**) Motility (percentage of non-paralyzed animals) of GMC101 treated with NV-128 at 150 μM; (**B**) Motility (percentage of non-paralyzed animals) of GMC101 treated with RKI-1447 at 20 μM; (**C**) Motility of GMC101 treated with clotrimazole at 0.1 μM; (**D**) Motility of GMC101 treated with midostaurin at 50 mM. Repetition of the experiments shown in Appendix A. Number of animals in each graph was at least 60.

**Figure 3 ijms-24-12079-f003:**
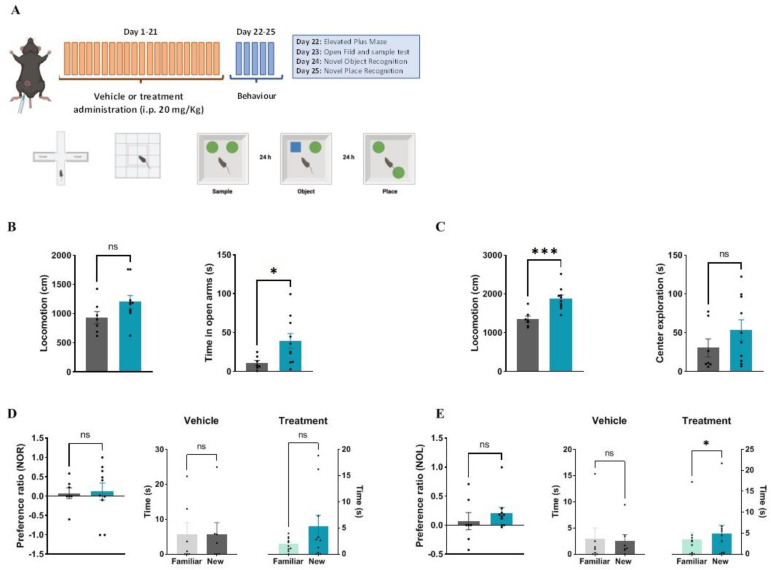
Effect of midostaurin on 3xTg mice behavior: (**A**) Experiment diagram. Green circles represent the objects that might be moved from the original place or replaced by a new object, represented as a blue square (**B**) Elevated plus maze test general motility results (left panel) and open-arm spent time (right panel) (gray bars represent averages of vehicle-treated mice and blue bars represent averages from midostaurin−treated mice); (**C**) Open-field maize general motility (left panel) and time spent in the center (right panel) (gray: vehicle, blue: midostaurin); (**D**) New object recognition assay results from left panel; untreated (gray bar) vs. treated (blue bar) mice. Central panel and right panel represent the time spent exploring the objects according to their treatment. (**E**) New-place recognition assay results from left panel; untreated (gray bar) vs. treated (blue bar) mice. Central panel and right panel represent the time spent exploring the objects according to their treatment. IP: intraperitoneally. * *p* < 0.05 and *** *p* < 0.001.

**Table 1 ijms-24-12079-t001:** Summary of the candidate genes identified by an eQTL in the central nervous system. GWAS and GTEx information included. Chr: chromosome. Ref: reference allele. Alt: alternative allele. O. R: odds ratio. ^a^: selected with different criteria (details further in this section). ^b^: disregarded in the drug selection. All these genes had a known ortholog in *C. elegans* according to the Ensembl database, except for PRKD3, KCNN4, and KCNH6.

SNP Information						GWAS		GTEx	
Gene Name	Tissue	rs	Chr	Ref	Alt	O. R	*p*-Value	Slope	Q-Value
** *CYP21A2* **	Putamen	rs1044506	6	T	G	1.12	3.49 × 10^−4^	0.66	1.87 × 10^−4^
** *DMPK* **	Cerebellum	rs2014576	19	G	A	1.10	7.32 × 10^−3^	0.22	4.26 × 10^−4^
** *EGFR* **	Caudate	rs149352678	7	C	T	0.86	2.59 × 10^−2^	−0.37	1.85 × 10^−3^
** *KCNH6* ** ** ^a^ **	Thyroid	rs1386502	17	T	C	0.92	1.13 × 10^−4^	−0.54	1.70 × 10^−3^
** *KCNN4* **	Substantia nigra	rs1386502	19	G	C	0.91	2.93 × 10^−5^	0.63	9.92 × 10^−4^
** *KCNN4* **	Amygdala	rs62116961 ^b^	19	T	C	0.92	1.13 × 10^−4^	−0.54	1.70 × 10^−3^
** *MARK3* **	Cortex	rs2296486	14	A	G	1.08	3.96 × 10^−4^	0.35	1.30 × 10^−3^
** *NDUFS2* **	Frontal cortex (BA9)	rs1136224	1	A	G	1.11	7.19 × 10^−4^	0.27	2.28 × 10^−4^
** *PRKD3* **	Putamen	rs2540974	2	A	G	0.91	1.12 × 10^−4^	−0.22	1.02 × 10^−4^
** *VKORC1* **	Cortex	rs881929	16	G	T	1.10	2.89 × 10^−2^	0.27	3.96 × 10^−4^

**Table 2 ijms-24-12079-t002:** Final drug selection for the validation assay in *C. elegans*. PCID: PubChem ID.

Gene	Drug	Interaction	PCID
** *DMPK* **	RKI-1447	Inhibitor	60138149
** *KCNH6* **	NS1643	Activator	10177784
** *KCNN4* **	Clotrimazole	Inhibitor	2812
** *NDUFS2* **	NV-128	Inhibitor	78357796
** *PRKD3* **	Midostaurin	Inhibitor	9829523

## Data Availability

The GTEx dataset was downloaded on 4 October 2021 from https://gtexportal.org/home/datasets (GTEx_Analysis_v8_eQTL.tar). Drug interaction information (interactions.tsv) was downloaded on 18 January 2022 from https://www.dgidb.org/downloads. The Alzforum Therapeutics database was downloaded on 1 July 2022 from https://www.alzforum.org/therapeutics, with Alzheimer’s Disease checked in the filters. The NIAGADS NG00067 list of AD loci and genes with genetic evidence compiled by the ADSP Gene Verification Committee was downloaded on 28 December 2022 from https://adsp.niagads.org/gvc-top-hits-list/ (Table 1 and Table 2). Code availability: DAGGER 0.7. The version used in this paper is publicly available at https://github.com/AEstebanMar/DAGGER-publication- (accessed on 22 April 2023).

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
