# Peer review of "A Functional Pipeline of Genome-Wide Association Data Leads to Midostaurin as a Repurposed Drug for Alzheimer’s Disease"

_ijms, 2023, doi:10.3390/ijms241512079_

Round 1
Reviewer 1 Report
The use of In silico drug discovery pipelines that utilize publicly available data and generate hypotheses that can be tested in cell culture and animal models is an important step in the drug discovery process for Alzheimer’s disease.
In the introduction please update lecanemab findings. I would additionally recommend citing more recent work questioning the efficacy of amyloid antibody drug interventions particularly the cognitive effects (below the minimum clinically important difference; MCID). The pipeline that you are suggesting begins with GWAS but then extends beyond that to test “top ranking” drugs in experimental models. I suggest indicating as such at the end of the introduction as this is a powerful validation/ translation part of your drug discovery model.
The drug discovery model is visualized in Figure 1 however the rationale is not clearly articulated. In the introduction you only describe the use of GWAS wheareas the process to identifying 22 drugable targets and ranking those targets includes steps beyond simply a GWAS analysis. This needs to be described clearly with rationale.
Some additionally challenges with the computational pipeline:
The first stage of the authors' novel DAGGER pipeline integrates genome-wide association study (GWAS) data with expression quantitative trait loci (eQTL) data and thus can be considered a novel approach to transcriptome-wide association studies (TWAS) and, more generally, to genomic fine mapping. In recent years several state-of-the-art computational approaches to TWAS were published (S-PrediXcan and S-MultiXcan by Hae Kyung Im's group, SMR/HEIDI by Jian Yang's group, and TWAS/FUSION and FOCUS by Bogdan Panasiuc's group) and have since been widely applied, also for AD (e.g Alzheimer's Research & Therapy 2020; 12:43). Compared to the author's DAGGER, these earlier techniques utilize more information contained in the GWAS dataset (all SNPs not just significantly associated ones under an arbitrary p-value threshold; moreover the SNPs' effect size and its statistical error). Also, unlike DAGGER, the earlier approaches take linkage disequilibrium (LD) into account in order (1) to associate GWAS SNPs with eQTL SNPs and (2) even to correct for LD-induced correlation of SNPs. These differences and limitations will need to be accounted for either in an updated/ modified pipeline or in the Discussion.
The second stage of the DAGGER pipeline attempts to identify drugs acting on the genes that the first, TWAS, stage of DAGGER detected as functionally linked to AD. Instead of taking a network pharmacology approach, the authors opt here for a simple approach that ignores potentially useful information in the network of gene-gene interactions. Again the authors’ response to this would be helpful and this is another limitation that would need to be discussed in the Discussion.
The identification of “top 5” targets suggests a ranking however this does not seem to be the case. As described in lines 118-126, its seems like the approach was based on multiple factors. It would be helpful (again as indicated previously) that the pipeline (not just the results) are described in a systematic way.
For the animal models, multiple cognitive/behavioral tests are presented. It would be helpful to indicate what the primary cognitive/behavioral test was as well as secondary tests. Similar to a clinical trial, if these are not indicated clearly (for example in a clearly articulated pipeline) then results can seem like cherry picking. Also, were there any blood collected for biomarker studies or postmortem path to look at impact on AD pathology?
Minor spelling errors throughout that need to by proofread
Author Response
In the introduction please update Lecanemab findings. I would additionally recommend citing more recent work questioning the efficacy of amyloid antibody drug interventions particularly the cognitive effects (below the minimum clinically important difference; MCID).
RE: We have modified the text according to reviewer’s suggestions and introduced references 9 to 12.
The pipeline that you are suggesting begins with GWAS but then extends beyond that to test “top ranking” drugs in experimental models. I suggest indicating as such at the end of the introduction as this is a powerful validation/ translation part of your drug discovery model. The drug discovery model is visualized in Figure 1 however the rationale is not clearly articulated. In the introduction you only describe the use of GWAS wheareas the process to identifying 22 drugable targets and ranking those targets includes steps beyond simply a GWAS analysis.This needs to be described clearly with rationale.
RE: The intrinsic problem of GWAS comes from multiple testing corrections. Nowadays thresholds are set around 10E-7, what leaves hundreds of false negatives. We tried to be more “generous” with the alpha-threshold and filter the results with the functional perspective, what also helps to model the pharmacological interactions that might be required (Figure 1). We tried to explain this along the text but we have rephrased some parts in order to make it clearer.
The first stage of the authors' novel DAGGER pipeline integrates genome-wide association study (GWAS) data with expression quantitative trait loci (eQTL) data and thus can be considered a novel approach to transcriptome-wide association studies (TWAS) and, more generally, to genomic fine mapping. In recent years several state-of-the-art computational approaches to TWAS were published (S-PrediXcan and S-MultiXcan by Hae Kyung Im's group, SMR/HEIDI by Jian Yang's group, and TWAS/FUSION and FOCUS by Bogdan Panasiuc's group) and have since been widely applied, also for AD (e.g Alzheimer's Research & Therapy 2020; 12:43). Compared to the author's DAGGER, these earlier techniques utilize more information contained in the GWAS dataset (all SNPs not just significantly associated ones under an arbitrary p-value threshold; moreover the SNPs' effect size and its statistical error). Also, unlike DAGGER, the earlier approaches take linkage disequilibrium (LD) into account in order (1) to associate GWAS SNPs with eQTL SNPs and (2) even to correct for LD-induced correlation of SNPs. These differences and limitations will need to be accounted for either in an updated/ modified pipeline or in the Discussion. The second stage of the DAGGER pipeline attempts to identify drugs acting on the genes that the first, TWAS, stage of DAGGER detected as functionally linked to AD. Instead of taking a network pharmacology approach, the authors opt here for a simple approach that ignores potentially useful information in the network of gene-gene interactions. Again the authors’ response to this would be helpful and this is another limitation that would need to be discussed in the Discussion.
RE: We would like to thank the reviewer for these helpful comments. It is true that we did not take LD into account. The rational of decision is that we were not seeking causative mutations, but rather analyzing the impact on transcription in order to model potential gene-drug interactions. We are aware that this implies a limitation in our analysis, as we have stated in the discussion. Regarding the network analysis the reviewer is again right,…we might be missing plenty of potential information of second-order and third-order of molecular interactions. The reason is that these network analysis are complex and exceed our know-how. We are trying to collaborate with a group whose IP is expert in modeling such interactions because the statistics below this modeling exceed our expertise.
In the R1 version of our maniscript we have included and discussed these items.
The identification of “top 5” targets suggests a ranking however this does not seem to be the case. As described in lines 118-126, its seems like the approach was based on multiple factors. It would be helpful (again as indicated previously) that the pipeline (not just the results) are described in a systematic way.
RE: Done as suggested. This stems from the same issue addressed by the aforementioned comments (pipeline beyond GWAS analysis).
For the animal models, multiple cognitive/behavioral tests are presented. It would be helpful to indicate what the primary cognitive/behavioral test was as well as secondary tests. Similar to a clinical trial, if these are not indicated clearly (for example in a clearly articulated pipeline) then results can seem like cherry picking. Also, were there any blood collected for biomarker studies or postmortem path to look at impact on AD pathology?
RE: We tried to present the mice behavioral analysis according to what previously published for this AD model. At this stage of our research there were no previous assumptions on what we were going to find. We could not anticipate whether the effect may apply to anxiety, place recognition, object recognition…for this reason the “whole package” was done, in order to validate our pipeline. In future works, we might perform a more profound characterization of the effect of Midostaurin, both at a behavioral and molecular level, as indicated by reviewer 1, however these experiments are costly and need a specific funding, missing at this moment.
Reviewer 2 Report
In the study by Esteban-martos et al., the authors use GWAS and Gene expression profiling as drug repositioning tool for Alzheimer’s disease. Based on the analysis, the authors narrow down 22 drug intervenable genes targets of which five genes were selected based on their expression profiles in the CNS. The author target the protein encoded by the genes using pharmacological interventions in a regulatable C. elegans model of Alzheimer’s disease expressing Amyloid-β and identify Midostaurin as potential candidate drug. Furthermore, the authors assess the effect of Midostaurin in Mice model of AD (3xTg model) and show behavioral improvements post drug treatment. Overall, the authors suggest that the DAGGER platform can be a reliable tool for drug repositioning in AD and other spectrum of genetic disorders.
I have the following comments:
The author does not mention or describe the term “Dagger” in the section “introduction’ which has to be briefly elaborated. The term should be removed from the title and re-phrased.
In the Method section: C. elegans motility assays: lines 214-215: The exact concentration of DMSO used must be provided. This section is particularly confusing. Why were the eggs incubated with different drugs when the purpose was to assess motility of the worms? This alters the parameters of the experiment, mainly time of drug exposure. The authors cite “Chen et al., 2015” as reference for the experimental procedure, which is methodologically different from what the authors perform in the study. In addition, information on how the paralyzed vs non-paralyzed worms were scored is missing.
Was Elisa or immunoblot analysis performed on the C. elegans model post induction at 25â—¦C to confirm the expression of β-amyloid in the experimental groups (in comparison to worms reared at 16â—¦C).
In discussion, the authors mention reproducible results. How many times each experiment was repeated and how many samples per group/experiment were used?
The authors performed behavioral analysis three weeks post-treatment using Midostaurin. Did the authors measure other parameters such as weight change, food / water consumption in the rodents during the period, and between the groups? If so, the results must be provided.
How were the graphs in figure 2 generated? Based on figure 2, the graph plot of control groups is different between 2A-B and 2C-D, which is odd. The same applies to control groups in the supplementary result section.
The legend for supplementary figures has to be included as separate supplementary file and not in the main manuscript since it leads to confusion. The legend for table S2 is missing.
There are several spelling, grammatical, and syntax errors in the manuscript and requires extensive revision for English language.
Author Response
In the study by Esteban-martos et al., the authors use GWAS and Gene expression profiling as drug repositioning tool for Alzheimer’s disease. Based on the analysis, the authors narrow down 22 drug intervenable genes targets of which five genes were selected based on their expression profiles in the CNS. The author target the protein encoded by the genes using pharmacological interventions in a regulatable C. elegans model of Alzheimer’s disease expressing Amyloid-β and identify Midostaurin as potential candidate drug. Furthermore, the authors assess the effect of Midostaurin in Mice model of AD (3xTg model) and show behavioral improvements post drug treatment. Overall, the authors suggest that the DAGGER platform can be a reliable tool for drug repositioning in AD and other spectrum of genetic disorders.
I have the following comments:
The author does not mention or describe the term “Dagger” in the section “introduction’ which has to be briefly elaborated. The term should be removed from the title and re-phrased.
RE: As suggested we have removed the name “DAGGER” from title. The term has been explained in abstract and introduction.
In the Method section: C. elegans motility assays: lines 214-215: The exact concentration of DMSO used must be provided. This section is particularly confusing. Why were the eggs incubated with different drugs when the purpose was to assess motility of the worms? This alters the parameters of the experiment, mainly time of drug exposure. The authors cite “Chen et al., 2015” as reference for the experimental procedure, which is methodologically different from what the authors perform in the study. In addition, information on how the paralyzed vs non-paralyzed worms were scored is missing.
RE: We are agree that the text was not clear. We have rewritten the paragraph in order to clarify it: “The different drugs were prepared at 100x the final concentration in 100% dimetil-sulfoxide (DMSO). Final concentrations in plate were: Midostaurin (50 mM), clotrimazol (0.1mM), NS1643 (30 mM), RKI-1447 20 mM and NV-128 150 mM (MedchemExress LLC, New Jersey, USA). As negative control 1% DMSO was used.”
Was Elisa or immunoblot analysis performed on the C. elegans model post induction at 25â—¦C to confirm the expression of β-amyloid in the experimental groups (in comparison to worms reared at 16â—¦C).
RE: The GMC101 strain is a well-stablished screening model and other authors already study that the expression of beta-amyloid and determine that the paralysis phenotype correlates with the muscle expression and aggregation of the beta-amyloid. For a more specific explanation we inserted reference 34: McColl G, et al Mol Neurodegeneration 2012 (https://doi.org/10.1186/1750-1326-7-57). We have rewritten the sentence to explain this fact.
In discussion, the authors mention reproducible results. How many times each experiment was repeated and how many samples per group/experiment were used?
RE: We have included in materials and methods section that the experiments were repeated at least twice.
The authors performed behavioral analysis three weeks post-treatment using Midostaurin. Did the authors measure other parameters such as weight change, food / water consumption in the rodents during the period, and between the groups? If so, the results must be provided.
RE: Animals underwent daily evaluations to assess their well-being and monitor any potential changes throughout the experiment. Every day, before and after administering the respective injection, careful monitoring was conducted on mice to observe signs of sickness. In this manner, a dishevelled coat, a downcast disposition, skin wounds, sluggishness, closed or narrowed eyes, hesitancy to move even when handled, and a decreased appetite were closely observed. Notably, no indications of illness were detected.
How were the graphs in figure 2 generated? Based on figure 2, the graph plot of control groups is different between 2A-B and 2C-D, which is odd. The same applies to control groups in the supplementary result section.
RE: For each graph we used 60 nematodes, so graphs show the behavior of the treated population vs non-treated. Drugs NV-128 and RKI-1447 were assayed simultaneously together with the same non-treated population as a control. Drugs Clotrimazole and Midostaruin were assayed in a different trial, using another non-treated population as control. For the supplementary figures, the experiments were performed in a similar way. We have change the figure legend to explain better this issue.
The legend for supplementary figures has to be included as separate supplementary file and not in the main manuscript since it leads to confusion. The legend for table S2 is missing.
RE: Done as suggested. Legends for supplementary figures have been moved from main text to a supplementary file. The missing legend for table S2 has been added.
Reviewer 3 Report
Esteban-Martos et al. present a research paper that outlines a drug repurposing method utilizing GWAS data in a rational manner. The approach, named as DAGGER, combines the assessment of SNP's Odds Ratio with tissue expression data sourced from GTEx. This methodology surpasses the examination of mutations in isolated genes and instead concentrates on the functional interplay between genomics and transcriptomics.
The paper is well-presented, well-developed, and well-written. However, I have some comments:
Introduction:
Lines 62-65: I suggest including a discussion on the limitations and challenges of GWAS studies to provide a balanced perspective. It is important to highlight that while GWAS is a powerful tool, it alone is not sufficient to fully understand genetic diseases, as previously believed. The DAGGER approach, as implemented by the authors, addresses this limitation by incorporating SNPs associated with changes in mRNA levels.
Results:
Lines 77, 126, and others: It appears that there is no supplemental information available to verify the information mentioned. This might not be due to the authors, but it is worth noting.
Line 130: Please provide an explanation of why GMC101 is considered a humanized C. elegans strain. Although I understand the reason, it may not be clear to all readers. Adding a reference to support this would be also beneficial.
Line 131: It would be helpful to include references for the previously reported clinical range concentrations as mentioned in the paper.
Figure 2:
- Only midostaurin improves the motility of GMC101, but only at certain concentrations. I have a question regarding the other three drugs: Would a higher concentration also lead to an improvement in motility? Does a higher concentration of midostaurin lead to an improved outcome?
- N ≥ 60, but only one worm from each condition is represented in the four graphs. Since the improvement in motility under this treatment with midostaurin is not substantial, it would be important to present the complete data with error bars and p-values.
- Are the control worms in panels A and B the same, as well as in panels C and D? It would be preferable to use different examples, considering there are more than 60 examples available. However, representing the average is still preferred.
Line 143: Why was a daily administration of 20 mg/kg chosen specifically? What was the rationale behind this choice and not any other dose?
Figure 3: Could you clarify the meaning of "familiar" and "new" in the context of the x-axis?
Discussion:
I would have appreciated a more in-depth discussion. Part of the Discussion is a summary of the followed protocol described earlier. It would be beneficial to explore the implications and broader significance of the findings in more detail.
Minor comments:
Line 39. Molecular biology.
Line 52. Aducanumab.
Line 120. PRKD3 in italics.
Line 128. C. elegans in italics.
Figure 2. Change uM for (micro)M.
Line 139. 0,1 (micro)M, not 0.1 (micro)M
Author Response
Esteban-Martos et al. present a research paper that outlines a drug repurposing method utilizing GWAS data in a rational manner. The approach, named as DAGGER, combines the assessment of SNP's Odds Ratio with tissue expression data sourced from GTEx. This methodology surpasses the examination of mutations in isolated genes and instead concentrates on the functional interplay between genomics and transcriptomics.
The paper is well-presented, well-developed, and well-written. However, I have some comments:
Introduction:
Lines 62-65: I suggest including a discussion on the limitations and challenges of GWAS studies to provide a balanced perspective. It is important to highlight that while GWAS is a powerful tool, it alone is not sufficient to fully understand genetic diseases, as previously believed. The DAGGER approach, as implemented by the authors, addresses this limitation by incorporating SNPs associated with changes in mRNA levels.
RE: Done as suggested. Limitations has been highlighted and addressed in lines 68-70
Results:
Lines 77, 126, and others: It appears that there is no supplemental information available to verify the information mentioned. This might not be due to the authors, but it is worth noting.
RE: Done as suggested. Within supplementary tables we have introduced a reference in line 91 and lines 101-102.
Line 130: Please provide an explanation of why GMC101 is considered a humanized C. elegans strain. Although I understand the reason, it may not be clear to all readers. Adding a reference to support this would be also beneficial.
RE Using the term “humanized” to reinforce the idea that the genomic cassette expresses the human form of Aß1-42 . However, it the reviewer or the editor considers it, we might change the term to “transgenic”. We have extended along the text our explanation on this model. “C. elegans GMC101 strain, where full length human Aß1-42 is expressed in muscles. This strain generates amyloid aggregates when incubated at 25ºC which correlates with paralysis [18]–[21]” And for those readers particularly interested on its screening capacity, its development, immunohistochemistry and molecular characterization, more information can be read on reference 34 [G. McColl et al., “Utility of an improved model of amyloid-beta (Aβ1-42) toxicity in Caenorhabditis elegans for drug screening for Alzheimer’s disease,” Mol Neurodegener, vol. 7, no. 1, p. 57, Nov. 2012,]
Line 131: It would be helpful to include references for the previously reported clinical range concentrations as mentioned in the paper.
RE: We have stated in the text that the “selected concentration ranges were chosen in accordance to the data available from the European Medical Agency”, however, citing a specific clinical trial would induce to misinterpretation. Any reader with expertise in oncology might be aware that doses largely vary if we talk about Acute leukemia or Non-small cell lung cancer. For this reason we included in table 2 the Pubchem ID where any reader have a direct summary not only to the clinical trials doses, but also to the ones used in cell culture and animal models for molecular characterization.
Figure 2: Only midostaurin improves the motility of GMC101, but only at certain concentrations. I have a question regarding the other three drugs: Would a higher concentration also lead to an improvement in motility? Does a higher concentration of midostaurin lead to an improved outcome?
RE: The GMC101 model were used as a cost-effective screening prior to the 3xtg mice. For this reason we used a single dose per candidate drug. The truly relevant given their potential extrapolation to humans are those concentration evaluated in mice…and these assays are largely conditioned by their cost. We chose the 20mg/kg according to the European Medical Agency and previous reports, and our data serve as a proof-of-concept that Midostaurin might be considered for repurposing for AD. However this opens its own research line that will require additional analysis, replication and fine-tune of the dosing, if funding is obtained.
N ≥ 60, but only one worm from each condition is represented in the four graphs. Since the improvement in motility under this treatment with midostaurin is not substantial, it would be important to present the complete data with error bars and p-values.
RE: We have included a graph with error bars and the p-value in the figure legend. Also a sentence explaining that for each graph we used at least 60 animals.
Are the control worms in panels A and B the same, as well as in panels C and D? It would be preferable to use different examples, considering there are more than 60 examples available. However, representing the average is still preferred.
RE: For each graph we used 60 nematodes, so the graph is showing the behavior of the populations treated vs non-treated. The drugs NV-128 and RKI-1447 were assayed simultaneously together with the same non-treated population as a control. The drugs Clotrimazole and Midostaruin were assayed in a different trial, using other non-treated population as control. We have change the figure legend to explain better this issue.
Line 143: Why was a daily administration of 20 mg/kg chosen specifically? What was the rationale behind this choice and not any other dose?
RE: As aforementioned, doses were selected after considering the available information not only from pubmed, but also from the European Medical Agency dossiers. Regarding Midostaurin, the 20 mg/kg dose was selected according to previous papers characterizing the effect of Modostaurin in rodent models (doi: 10.3390/biom11070972) and EMA dossier of “Rydapt”, the commercial name of Midostaurin: https://www.ema.europa.eu/en/documents/assessment-report/rydapt-epar-public-assessment-report_en.pdf
Figure 3: Could you clarify the meaning of "familiar" and "new" in the context of the x-axis?
RE: The recognition tests measure the time spend exploring familiar and new objects. Cognitively healthy mice spend more time around new objects than sick (cognitively compromised) mice. We explain this in section 4.5. Behavioural testing:
“Briefly, two identical copies of an object (i.e. 'familiar' object) were placed in the apparatus near two adjacent corners, and mice were left to explore for 10 min (sample session). On day 24, mice were allowed to explore for 10 min an identical copy of the familiar object and a 'novel' unknown object located in one of the previous positions (object memory test). The preference ratio (%) [(time exploring the novel object−time exploring the familiar object)/total time exploring both objects] and time spent exploring both objects were determined. Finally, on day 25, the novel place recognition (NPR) test was performed. Mice were allowed to explore for 10 min two identical copies of the familiar object, one placed in its habitual position and the other displaced to an opposite corner (place memory test). The preference ratio (%) [(time exploring the displaced object−time exploring the static object)/total time exploring both objects] and total time exploring the familiar and displaced objects were determined.”
Author Response
lines 199-200: pval and Qcutoff are parameter names in the code. “pval” can be easily understood as a p-value setting, because it is a common name among other softwares too, but not Qcutoff. I suggest “We used the options with p-value of 0.001 and the cutoff value for Q (I don’t know what Q stands for, so please explain) of 5” instead.
RE: Done as suggested, along lines 213-217
line 202: 50 % -> 50%
RE: Done as suggested.
line 250: Data were analyzed for conditions of normality with -> “We tested normality of data using”
RE: Done as suggested.
line 250: Grubbs’ test was applied to identify outliers -> “and outliers were checked by Grubb’s test”
RE: Changed as suggested
line 251: used together – do you mean they were used as two-sample independent test, or paired test? (It can’t be a paired test, so you should simply drop the word “together”)
RE: Changed as suggested.
line 252: single-factor – do you mean one variable of interest? “Factor” is used only when you ran the factor analysis. I do suggest re-wording this section.
RE: Corrected, factor -> variable
line 255: Data are expressed as -> what do you mean by this? Data were obtained as they were.
RE: Corrected, expressed -> represented
line 255: SEM -> margin of error, which is critical value * SE.
RE: Corrected, SEM -> SE.
Minor
I don’t know if this format was required by the journal, but it looked strange to me:
Subsection titles are in bold while section titles are not. Materials and Methods section comes after Discussion section. (Usually MM section comes right after the introduction, before results) If were not required so, please update accordingly.
RE: Thanks for your comment. We have followed the Journal’s guidelines for the submission, although we agree that its quite different from other editorials.
I would have appreciated a more in-depth discussion. Part of the Discussion is a summary of the followed protocol described earlier. It would be beneficial to explore the implications and broader significance of the findings in more detail.
RE: We should stress that our work describes a novel methodology, a pipeline that can be applied to different GWAS data, maybe that’s the reason we recapitulate the workflow at the end of the manuscript. Nevertheless we have followed reviewer’s suggestion and made it more extense (lines 199-202).
Minor comments:
Line 39. Molecular biology.
RE: Corrected as suggested.
Line 52. Aducanumab.
RE: Corrected as suggested.
Line 120. PRKD3 in italics.
RE: Corrected as suggested.
Line 128. C. elegans in italics.
RE: Corrected as suggested.
Figure 2. Change uM for (micro)M.
Line 139. 0,1 (micro)M, not 0.1 (micro)M
RE: Corrected as suggested
Round 2
Reviewer 2 Report
The authors have successfully addressed the issues raised by this reviewer.
The quality of language has improved. There are still a very few grammatical errors in the manuscript.
Author Response
On behalf of all the authors, I would like to express our gratitude to Reviewer 2 for his/her contributions in enhancing our manuscript
Reviewer 3 Report
Thanks for the comments. I support publication.
Author Response
On behalf of all the authors, I would like to express our gratitude to Reviewer 3 for huis/her contributions in enhancing our manuscript
Reviewer 4 Report

mentioned in my review
Author Response
Normality, outliers were tested. But I couldn’t find any paragraph throughout the manuscript, mentioning or summarizing the findings, e.g. all variables were normally distributed, there was no outliers, there were outliers as *** and we transformed as ***. Also, why did you check normality? Why did you check for outliers? There are reasons for doing these (you don’t have to mention why, in the manuscript) but the findings should be mentioned as a preliminary work.
RE: Normality determination was not a result per se. It was done in order to decide for the most appropriate statistical test to use. For non-Normally distributed variables, non-parametric tests were used. Parametric tests were used only when Nornality was comfirmed. We have rephrased the paragraph as follows:
“We tested normality of data using Kolmogorov-Smirnov's test, and outliers were checked by Grubb’s test. Data from female and male mice were used for analysis, as no significant differences between sexes were found by Student’s t-test. For the normal single-variable case, we analyzed data using two-tailed Student's t-test, and two-tailed Mann-Whitney test for data with non-normal unpaired distribution; or Wilcoxon test for paired non-normal distribution data.”
Regarding the search for outliers, we proceeded in this way different reasons, and according to previously published recommendations:
Outliers can significantly affect the results of statistical analyses, leading to misleading conclusions. Outliers may arise due to random errors, measurement issues, uncontrolled infections or other anomalies. By eliminating outliers, we aim to improve the overall data quality and enhance the trustworthiness of the findings.
- Aggarwal, C. C. (2016). Outlier Analysis. Springer International Publishing.
- Fox, J., & Weisberg, S. (2019). An R Companion to Applied Regression. Sage Publications Ltd.
- Wilcox, R. R. (2017). Introduction to Robust Estimation and Hypothesis Testing. Academic Press.
Lines 290-292: the statement is vague.
RE: The paragraph has been modified.
Line 292: “C. elegans motility analysis was plot using ...” - do you mean “was plotted” ?
RE: Thanks for the comment. Corrected as indicated.